# Association Between Active DNA Demethylation and Liver Fibrosis in Individuals with Metabolic-Associated Steatotic Liver Disease (MASLD)

**DOI:** 10.3390/ijms26031271

**Published:** 2025-01-31

**Authors:** Ilaria Barchetta, Michele Zampieri, Flavia Agata Cimini, Sara Dule, Federica Sentinelli, Giulia Passarella, Alessandro Oldani, Katsiaryna Karpach, Maria Giulia Bacalini, Marco Giorgio Baroni, Anna Reale, Maria Gisella Cavallo

**Affiliations:** 1Department of Experimental Medicine, Sapienza University of Rome, 00161 Rome, Italy; ilaria.barchetta@uniroma1.it (I.B.); michele.zampieri@uniroma1.it (M.Z.); flaviaagata.cimini@uniroma1.it (F.A.C.); sara.dule@uniroma1.it (S.D.); giulia.passarella@uniroma1.it (G.P.); alessandro.oldani@uniroma1.it (A.O.); katsiaryna.karpach@uniroma1.it (K.K.); anna.reale@uniroma1.it (A.R.); 2Endocrinology and Diabetes, Department of Clinical Medicine, Public Health, Life and Environmental Sciences (MeSVA), University of L’Aquila, 67100 L’Aquila, Italy; federica.sentinelli@univaq.it (F.S.); marcogiorgio.baroni@univaq.it (M.G.B.); 3IRCCS Istituto delle Scienze Neurologiche di Bologna, 40139 Bologna, Italy; mariagiuli.bacalini2@unibo.it; 4Neuroendocrinology and Metabolic Diseases, IRCCS Neuromed, 86077 Pozzilli, Italy

**Keywords:** epigenetics, DNA methylation, DNA demethylation, non-alcoholic fatty liver disease, metabolic-associated fatty liver disease, NASH, liver fibrosis, fibrosis NASH index, FNI

## Abstract

Metabolic-associated steatotic liver disease (MASLD) represents the most common chronic hepatopathy worldwide and an independent risk factor for cardiovascular disease and mortality, particularly when liver fibrosis occurs. Epigenetic alterations, such as DNA methylation, may influence MASLD susceptibility and progression; yet mechanisms underlying this process are limited. This study aimed to investigate whether active DNA demethylation in peripheral blood mononuclear cells (PBMCs) from individuals with MASLD, alongside the methylation and mRNA levels of inflammation- and fibrosis-related candidate genes, is associated with liver fibrosis. For this study, global demethylation intermediates (5-hydroxymethylcytosine [5hmC], 5-formylcytosine [5fC]) were quantified in PBMCs from 89 individuals with/without MASLD using ELISA. Site-specific DNA methylation of SOCS3, SREBF1, and TXNIP was analyzed by mass spectrometry-based bisulfite sequencing; mRNA expression was assessed via RT-PCR. Individuals with MASLD and moderate-to-high fibrosis risk (estimated by the fibrosis non-alcoholic steatohepatitis (NASH) index, FNI) progressively exhibited greater global 5hmC and 5fC levels. Higher FNI was associated with reduced methylation of the SOCS3 gene and increased mRNA expression of the SOCS3, TXNIP, IL-6, and MCP-1 genes. In conclusion, elevated fibrosis risk in MASLD is associated with active global DNA demethylation, as well as differential methylation and expression patterns of genes, which are key regulators of inflammation and fibrosis. These epigenetic alterations in PBMCs may mirror DNA methylation changes in the liver, which may potentially contribute to liver fibrogenesis and represent novel biomarkers for MASLD progression toward fibrosis.

## 1. Introduction

Metabolic dysfunction-associated steatotic liver disease (MASLD) is the most common chronic liver disease worldwide, affecting over 32% of adults globally, and is a critical hepatic component of metabolic dysfunction, closely related to conditions such as obesity, type 2 diabetes mellitus (T2DM), hypertension, and dyslipidemia [1,2]. Indeed, the nomenclature MASLD has been recently adopted to emphasize the strong metabolic basis of hepatic steatosis, replacing the older term non-alcoholic fatty liver disease (NAFLD) [3]. MASLD represents an independent risk factor for overall mortality and cardiovascular disease and is associated with T2DM onset and detrimental outcomes, chronic kidney disease, and liver-specific outcomes like fibrosis and hepatocellular carcinoma [4]. Liver fibrosis, in particular, emerges as the principal determinant of morbidity and mortality in MASLD, with advanced fibrosis strongly associated with liver-related and cardio-metabolic complications [5,6]. Yet mechanisms driving MASLD progression towards fibrosis remain not completely understood [6]. In this context, identifying individuals at higher risk for fibrosis progression is a major clinical goal, as timely intervention can modify the natural progression of the disease and improve its outcomes, necessitating further research to optimize patient stratification and management strategies.

DNA methylation changes play a pivotal role in the pathogenesis and progression of metabolic diseases, serving as a critical interface between genetic predispositions and environmental influences [7]. Recent studies have emphasized the role of DNA methylation in the development and progression of MASLD [8].

The methylation process involves the enzymatic addition of a methyl group predominantly to cytosine residues followed by guanine, forming CpG dinucleotides in DNA and resulting in the formation of 5-methylcytosine (5mC). This modification silences genes by restricting chromatin accessibility and preventing transcription factor binding. DNA methylation is reversible through passive and active demethylation mechanisms. Passive demethylation primarily occurs during DNA replication, leading to a dilution of methylation signals. In contrast, active demethylation involves the enzymatic oxidation of 5mC into intermediates, including 5-hydroxymethylcytosine (5hmC), 5-formylcytosine (5fC), and 5-carboxycytosine (5caC).

Aberrant DNA methylation patterns are implicated in key aspects of metabolic diseases. In MASLD, differential DNA methylation profiles have been strongly associated with liver fibrosis severity [9]. Similarly, in T2DM, methylation changes have been linked to critical processes such as insulin resistance, systemic inflammation, and oxidative stress [10].

The precise mechanisms driving these methylation alterations remain unclear. Emerging evidence highlights the active demethylation pathway as a significant contributor to metabolic disease progression, including diabetes. Specifically, studies in animal models and preliminary experimental findings in humans suggest that active demethylation intermediates, such as 5hmC and 5fC, may serve as biomarkers of disease states and play a functional role in mediating disease-related epigenetic changes [11,12,13]. Conversely, mechanisms behind epigenetic modifications and differential methylation patterns in MASLD are not fully elucidated yet, and data on their association with liver fibrosis are limited. Thus, we hypothesize that changes in methylation profiles in PBMCs from patients with MASLD mirror the DNA active demethylation processes reported to drive liver fibrosis.

This study aimed to investigate in PBMCs the active DNA demethylation process in relation to MASLD-associated fibrosis risk by analyzing both global levels of the demethylation intermediates 5hmC and 5fC and exploring the methylation profiles and expression of suppressor of cytokine signaling 3 (SOCS3), sterol regulatory element binding transcription factor 1 (SREBF1), and thioredoxin interacting protein (TXNIP), key genes involved in inflammatory processes and liver fibrosis [14,15,16,17,18,19,20]. 

## 2. Results

Among 89 study participants, 63% (56/89) had a clinical diagnosis of MASLD according to the hepatic steatosis index (HSI > 36) calculation. These individuals had greater prevalence of T2DM and worse metabolic profile than those without MASLD, whereas age and sex distribution were comparable between sub-groups. Clinical characteristics of the study population according to the MASLD presence are reported in Table 1.

When dividing the study cohort in relation to the liver fibrosis risk, as estimated by FNI, we found that 44% of participants (39/89) were at moderate-to-high risk of hepatic fibrosis. 

In subjects at moderate-to-high fibrosis risk, the prevalence of male sex (74% vs. 50%, *p* = 0.02) and T2DM (95% vs. 38%, *p* < 0.001) was significantly higher than in those in the low-FNI group. Study participants at progressively higher risk of liver fibrosis had significantly increased BMI, FBG, blood transaminases, and lipids. In line with these findings, the prevalence of T2DM was also progressively higher, up to 100% among individuals with the greatest fibrosis risk (FNI ≥ 0.33). Clinical characteristics of the study population according to liver fibrosis classes are reported in Table 2.

### 2.1. Global Demethylation

#### 2.1.1. Demethylation Intermediates in Relation to MASLD

In our population, the presence of MASLD was associated with higher levels of the global demethylation intermediate 5hmC in DNA from PBMCs compared to non-MALSD (mean ± SD 5hmC: 0.076 ± 0.04 vs. 0.048 ± 0.04, *p* = 0.011; Figure 1). No difference was observed for 5fC between these subgroups (MASLD mean ± SD 5fC: 0.0055 ± 0.006 vs. non-MASLD: 0.0053 ± 0.004, *p* = 0.91).

#### 2.1.2. Demethylation Intermediates and Liver Fibrosis

Subsequently, in order to explore the association between differential demethylation profiles and liver fibrosis risk, the same cohort (n = 89) was stratified based on the risk of liver fibrosis, estimated by FNI. Indeed, individuals at higher risk of liver fibrosis exhibited significantly higher levels of global DNA active demethylation intermediates in PBMCs, compared to both those at moderate and low risk, as indicated by significantly greater global levels of 5hmC (mean ± SD 5hmC in low risk: 0.048 ± 0.02, moderate risk: 0.051 ± 0.02, high risk: 0.10 ± 0.03, *p* = 0.003; one-way ANOVA test; Bonferroni’s adjustment: high vs. moderate risk: *p* = 0002; high vs. low risk: *p* < 0001) and 5fC (mean ± SD 5hmC in low risk: 0.005 ± 0.007, moderate risk: 0.003 ± 0.002, high risk: 0.010 ± 0.004, *p* = 0.007, one-way ANOVA test; Bonferroni’s adjustment: high vs. moderate risk: *p* = 0003; high vs. low risk: *p* < 0001; Figure 2). 

The significant association between higher DNA demethylation levels in PBMCs and more advanced fibrosis persisted when the analyses were performed in the subgroup of individuals with MASLD (HSI > 36). Among these patients, 64% had moderate-to-high fibrosis risk; 5hmC and 5fC progressively increased across classes at more pronounced fibrosis risk (5fC (mean ± SD) in low fibrosis risk: 0.0028 ± 0.001, moderate fibrosis risk: 0.0027 ± 0.002, high fibrosis risk: 0.0085 ± 0.003, *p* = 0.024) and 5hmC (mean ± SD in low fibrosis risk: 0.042 ± 0.023, moderate fibrosis risk: 0.039 ± 0.023, high fibrosis risk: 0.11 ± 0.027, *p* = 0.002).

At the bivariate analysis, the FNI, considered as a continuous variable, linearly correlated with both 5hmC and 5fC levels on DNA from PBMCs (r = 0.65, *p* < 0.001; r = 0.43, *p* = 0.016, respectively; Pearson’s coefficients; Appendix A). These associations persisted statistically significant after adjusting for sex (5hmC: r = 0.64, *p* < 0.001; 5fC: r = 0.61, *p* < 0.001) and for sex and presence of T2DM (5hmC: r = 0.63, *p* = 0.002; 5fC: r = 0.59, *p* = 0.003) at the partial correlation analysis. Conversely, no association was found between the global methylation profile, as indicated by 5-methylcytosine (5mC) levels, and either the presence of MASLD (mean ± SD 5mC MASLD: 0.71 ± 0.24, non-MASLD: 0.59 ± 0.22, *p* = 0.13) or fibrosis classes (mean ± SD 5mC in low risk: 0.53 ± 0.22, moderate risk: 0.80 ± 0.32, high risk: 0.69 ± 0.22, *p* = 0.10).

### 2.2. Gene-Specific Methylation and Associated mRNA Expression Levels

We specifically explored in PBMCs the DNA methylation levels of candidate genes involved in inflammatory processes and fibrogenesis, such as SOCS3, SREBF1, and TXNIP. The profiling was focused on regions previously associated with methylation changes in population studies conducted on whole blood [21,22]. In order to explore the existence of differential DNA methylation between subgroups of individuals at different fibrosis risk, we compared the CpG methylation levels between individuals stratified based on risk of liver fibrosis. Appendix A illustrates details of the genomic locations of the analyzed CpG sites within the genes, as well as their methylation levels, compared between individuals with low and moderate-to-high FNI scores. Subjects at moderate-to-high fibrosis risk had significantly lower methylation levels at specific CpG sites within the SOCS3 gene locus compared to patients at low fibrosis risk (CpG8: 0.060 ± 0.02 vs. 0.083 ± 0.037, *p* = 0.02; CpG13: 0.015 ± 0.02 vs. 0.037 ± 0.03, *p* = 0.01; CpG15.16: 0.54 ± 0.09 vs. 0.62 ± 0.1, *p* = 0.03; CpG17.18: 0.55 ± 0.12 vs. 0.63 ± 0.09, *p* = 0.03, Appendix A). 

The Pearson correlation analysis confirmed a negative association between FNI and the methylation levels of CpG13, CpG15.16, and CpG17.18 (CpG13: r = −0.35, *p* = 0.04; CpG15.16: r = −0.32, *p* = 0.012; CpG17.18: r = −0.25, *p* = 0.049; Appendix A). In contrast, the previously suggested association between CpG8 methylation and FNI, identified through stratification analysis, was not supported (r = −0.21, *p* = 0.10), possibly due to a non-linear relationship. Additionally, correlation analysis revealed further significant negative linear associations between FNI and the methylation levels of CpG5, CpG6, CpG20, and CpG28 (CpG5: r = −0.62, *p* = 0.032; CpG6: r = −0.35, *p* = 0.04; CpG20: r = −0.35, *p* = 0.04; CpG28: r = −0.26, *p* = 0.04; Appendix A). 

In line with the hypomethylation observed at multiple CpG sites within the SOCS3 gene, we identified a positive association between FNI and SOCS3 mRNA expression levels (r = 0.37, *p* = 0.017; Appendix A).

Notably, the mRNA expression of IL-6 and MCP-1 genes, which are entangled in pathways associated with SOCS3 activation, was progressively higher in PBMCs from individuals with more elevated estimated fibrosis risk (Figure 3). In addition, the FNI linearly correlated with their mRNA expression levels (IL-6: r = 0.47, *p* = 0.004; and MCP-1: r = 0.48, *p* = 0.002; Pearson’s coefficients, Appendix A).

Finally, the FNI was associated with TXNIP mRNA expression (r = 0.51, *p* = 0.002, Appendix A) and with the methylation levels of the SREBF1 gene at site CpG1 (r = −0.26, *p* = 0.04) and CpG4 (r = 0.34, *p* = 0.005; Appendix A).

## 3. Discussion

In this study, we examined the active DNA demethylation processes in relation to MASLD and associated liver fibrosis, revealing a significant increase in demethylation intermediates in PBMCs from individuals with MASLD. Additionally, in these patients, global levels of 5hmC and 5fC were positively correlated with the risk of liver fibrosis in a dose-dependent manner. These results further support the hypothesis that blood-based epigenetic markers mirror distinct DNA methylation patterns in MASLD, reflecting specific molecular pathways underlying the condition [8]. Specifically, differential DNA methylation profiles in MASLD involve key pathways related to insulin signaling, fibrosis, and metabolic dysfunction. Hypermethylation of genes associated with insulin signaling, focal adhesion, and extracellular matrix remodeling has been observed. These findings align with the primary pathogenic factor of MASLD—insulin resistance—which in turn triggers processes leading from hepatic steatosis to NASH and fibrosis development [23].

Hypomethylation, however, appears to be the predominant alteration in MASLD [24,25,26,27], affecting specific genes and repetitive sequences with implications for tissue repair, energy metabolism, and carcinogenesis in the liver. In animal models, the deprivation of methyl group donors or impairment of their metabolism can induce hepatic steatosis, while their supplementation can protect against the disease [28,29]. More recently, in animal and cellular models of hepatic steatosis, the disease was shown to be associated with genomic remodeling of 5hmC marks in a discrete set of functionally relevant genes, with implications for lipid metabolism, fibrosis initiation, and progression [30,31,32]. Notably, there is an overlap between methylation changes associated with MASLD observed in the liver and those identified in the blood, both reflecting increased inflammation [25]. This suggests that analyzing blood methylation changes may serve as a non-invasive marker of the disease.

Building on these premises, this study is the first to investigate the association between global and site-specific markers of active DNA demethylation in PBMCs and liver fibrosis risk in MASLD within a clinical context. Our data indicate a strong association between an increased risk of fibrosis, as measured by FNI, and elevated global levels of 5hmC and 5fC in PBMCs. These global markers of active demethylation were significantly correlated with higher FNI scores. These findings suggest that active demethylation processes may represent a key mechanism in MASLD progression, particularly in patients at higher risk of developing fibrosis. This observation aligns with previous studies showing that lower methylation levels are associated with higher degrees of liver inflammation, disease progression, and greater histological severity in NAFLD [33]. In our study, 5hmC levels were elevated in the presence of MASLD regardless of fibrosis risk, while 5fC levels showed a progressive increase across fibrosis risk classes. These findings underscore the potential of 5fC as a specific biomarker for identifying patients at high risk of advanced liver fibrosis. Interestingly, no association was found between global DNA methylation, measured by 5-methylcytosine (5mC) levels, and the presence of MASLD or fibrosis. This result suggests that site-specific active demethylation, rather than global demethylation, may contribute to MASLD progression. Indeed, while 5hmC is narrowly distributed across the genome, predominantly in genes and regulatory regions, most of the 5mC signal is localized in repetitive DNA regions. Consequently, a localized reduction in 5mC due to 5hmC formation does not necessarily result in a global decrease in 5mC.

A more detailed analysis of specific genes involved in inflammation and fibrogenesis revealed insights into the molecular mechanisms driving MASLD progression. We focused on SOCS3, SREBF1, and TXNIP, genes known to be implicated in inflammatory pathways and liver fibrosis [14,15,16,17,18,19,20]. Although these genes are not recognized as primarily fibrosis-specific genes, several recent studies have demonstrated that they centrally take part in fibrosis development and progression in several tissues and organs, such as the skin, liver, and kidney [18,19,20], and their epigenetic modulation could impact fibrogenesis in the experimental setting [18]. Thus, we found that PBMCs from individuals at moderate-to-high risk of liver fibrosis exhibited significantly lower DNA methylation levels at specific CpG sites within the SOCS3 gene locus compared to PBMCs from subjects at low risk of fibrosis. This suggests that reduced methylation at these sites may lead to the overexpression of SOCS3, which is involved in regulating inflammation and fibrosis [14,15]. Moreover, the inverse correlation between FNI and methylation at several CpG sites within the SOCS3 gene (e.g., CpG5: r = −0.62, *p* = 0.032) supports the notion that hypomethylation of SOCS3 is associated with higher fibrosis risk. In agreement with this, we also observed a positive association between FNI and the mRNA expression levels of SOCS3 (r = 0.37, *p* = 0.017). This finding highlights the potential role of SOCS3 as a molecular mediator of fibrogenesis in MASLD and suggests that its hypomethylation could be an important epigenetic alteration driving disease progression. Indeed, recent observations suggest that the SOCS3-JAK-STAT3 signaling pathway, activated by cytokines, plays a significant role in MASLD by promoting lipid accumulation and the infiltration of neutrophils and macrophages into the liver, increasing inflammation [34]. 

While the EpiTyper method is a robust tool for analyzing DNA methylation, its sensitivity threshold of detecting at least 5% differential methylation must be considered when interpreting results. In particular, the observed methylation differences at CpG8 and CpG13 within the SOCS3 gene fall close to this sensitivity limit. Therefore, these findings should be taken with caution, and additional validation using more sensitive methods would be beneficial to confirm their biological significance.

The origin of the observed methylation changes is difficult to determine, as DNA methylation is influenced by both environmental and genetic factors. Recent studies suggest that epigenetic alterations in MASLD are partially regulated by DNMT and TET enzymes, which are responsible for the deposition and removal of 5mC, respectively [30,35,36]. However, further research is needed to better understand their role in MASLD progression.

Additionally, mRNA expression levels of IL-6 and MCP-1, regulated by SOCS3, were progressively higher in PBMCs from individuals with greater fibrosis risk. This is in line with the evidence of the central role of inflammatory pathways in liver fibrosis progression in MASLD and suggests that epigenetic changes affecting SOCS3 could impact the expression of cytokines primarily involved in inflammation and fibrogenesis, such as IL-6 and MCP-1.

Finally, our results indicate that mRNA expression of TXNIP, a gene involved in oxidative stress and inflammation, was positively correlated with fibrosis risk, underscoring the role of oxidative stress in liver damage. Supporting this, Guo et al. [37] reported an association between elevated circulating TXNIP levels and fatty liver in patients with newly diagnosed T2DM, suggesting that TXNIP activation may contribute to metabolic liver disease development and progression through oxidative stress pathways.

Notably, some associations were detected exclusively through stratification, while others were identified only by linear correlation analysis. These discrepancies may be due to the differential sensitivity of statistical methods, with stratification potentially capturing threshold effects or categorical changes, while linear correlation is more suited to continuous, dose-dependent relationships.

The overall study results provide new insights into the contribution of epigenetic hallmarks in MASLD and show a specific differential demethylation pattern in relation to liver fibrosis. Although our study has a cross-sectional design, which does not allow for establishing a causal nexus between epigenetic changes and MASLD development/progression, it is plausible to hypothesize that the epigenetic modulation of genes involved in inflammation and fibrogenesis might, at least in part, contribute to the progression of liver damage once MASLD is established. The existence of a dose-dependent relationship between the fibrosis degree and both global active demethylation and site-specific methylation levels of candidate genes in PBMCs, along with the mRNA expression of pro-inflammatory cytokines, reinforces the hypothesis of a mechanistic involvement of epigenetic processes in MASLD onset and progression in the course of metabolic disorders. Furthermore, these correlations suggest a potential role of differential methylation patterns in disease screening and follow-up in MASLD, serving therefore as novel biomarkers in populations at high risk. Further studies are warranted in a longitudinal setting and across diverse populations, particularly those with more advanced liver disease and histologically confirmed liver fibrosis. 

The interpretation of our results must account for the limitation introduced by using PBMCs, as this approach cannot entirely exclude the potential influence of changes in cell composition on the observed methylation variations. Moreover, while studies such as Johnson et al. [25] have demonstrated that methylation profiles from blood samples can partially capture liver-specific patterns, indicating shared pathways in disease progression, substantial challenges persist in establishing the relevance of our blood-based findings to liver-specific epigenetic changes caused by the disease. Future research should prioritize integrating single-cell approaches or comparing matched primary hepatocytes and blood samples to better delineate tissue-specific epigenetic signatures and their clinical significance.

In conclusion, our study demonstrates for the first time the presence of differential global active DNA demethylation in MASLD, paralleled by changes in methylation and mRNA expression patterns of candidate genes associated with inflammation and fibrogenesis. These findings add knowledge on potential mechanistic processes behind fibrosis development and poor clinical outcomes in MASLD, which could be potentially addressed in a multi-targeted approach to MASLD treatment. Furthermore, identifying individuals at higher risk of disease progression could enhance risk stratification, enabling earlier intervention strategies aimed at modifying the course of the disease, finally improving clinical outcomes in MASLD.

## 4. Materials and Methods

### 4.1. Study Population 

For this cross-sectional investigation, we recruited 89 individuals (male/female sex: 32/57; mean ± SD age: 62.3 ± 10.5 years, mean ± SD body mass index (BMI) 28.5 ± 3.6 kg/m²) attending the Diabetes and Endocrinology outpatient clinic at Sapienza University, Rome, Italy, for metabolic evaluations and/or diabetes management. The inclusion/exclusion criteria for this study were male or female sex, age between 20 and 65 years; no history of excessive alcohol consumption (defined as daily alcohol intake exceeding 30 g for men and 20 g for women); negative results for hepatitis B surface antigen and hepatitis C virus antibody; no history of cirrhosis or other liver diseases (such as hemochromatosis, autoimmune hepatitis, or Wilson’s disease); no current treatment with medications known to induce liver steatosis (e.g., corticosteroids, estrogens, methotrexate, tetracycline, calcium channel blockers, or amiodarone). All study participants underwent medical history collection and clinical evaluations. Individuals without a confirmed diagnosis of T2DM underwent a standard oral glucose tolerance test (OGTT). Anthropometric measurements included weight and height, recorded with light clothing and without shoes, to calculate BMI (kg/m²). Waist circumference (in centimeters) was measured at the midpoint between the 12th rib and the iliac crest. Systemic systolic (SBP, mmHg) and diastolic (DBP, mmHg) blood pressure were measured after a 5-minute rest period, with three readings taken and the average of the second and third used in the analysis. Fasting blood samples were collected from all participants after an overnight fast of at least 8 h. The following biochemical parameters were measured: fasting blood glucose (FBG, mg/dL), glycated hemoglobin (HbA1c, %), aspartate aminotransferase (AST, IU/L), alanine aminotransferase (ALT, IU/L), gamma-glutamyl transferase (GGT, IU/L), total bilirubin (mg/dL), direct bilirubin (mg/dL), total cholesterol (mg/dL), high-density lipoprotein cholesterol (HDL, mg/dL), low-density lipoprotein cholesterol (LDL, mg/dL), triglycerides (mg/dL), and uric acid (mg/dL). The presence of ongoing chronic treatments was recorded. 

The presence of MASLD was diagnosed by clinical criteria [3] and then the hepatic steatosis index (HSI) was calculated according to the following formula: HSI = 8 × AST/ALT + BMI + (+2 if female) + (+2 if diabetes is present) [38]. The risk of liver fibrosis was estimated by calculating the fibrotic NASH index (FNI), a recently validated non-invasive scoring system used to assess the risk of advanced liver fibrosis in patients at high risk of non-alcoholic steatohepatitis (NASH). FNI was shown to perform better than other non-invasive indexes, such as FIB-4, in identifying liver fibrosis in individuals with metabolic disease and type 2 diabetes [39]. The formula for the FNI calculation includes AST, HDL cholesterol, and HbA1c (calculator available online at https://fniscore.github.io/).

### 4.2. Peripheral Blood Mononuclear Cells Isolation

Peripheral venous blood samples (6 mL) were drawn into EDTA-containing vacutainer tubes (BD). Peripheral blood mononuclear cells (PBMCs) were separated by density gradient centrifugation using the Lymphoprep™ solution (density: 1.077 g/mL; Axis-Shield, Dundee, UK), following the manufacturer’s protocol. The isolated cells were either processed immediately or stored as pellets at −80 °C for later use.

### 4.3. Extraction of Total DNA and Quantification of Global 5mC, 5hmC, and 5fC Levels

Total DNA was extracted from PBMCs using the DNeasy Blood and Tissue Kit (Qiagen, Milan, Italy) in accordance with the manufacturer’s guidelines. The purified DNA was utilized to measure the global levels of 5mC, 5hmC, and 5fC using specific quantification kits: MethylFlash Methylated DNA Quantification Kit (Epigentek, New York, NY, USA) for 5mC, MethylFlash Hydroxymethylated DNA Quantification Kit (Epigentek, NY, USA) for 5hmC, and MethylFlash 5-Formylcytosine DNA Quantification Kit Epigentek, NY, USA) for 5fC. Chemiluminescent signals were detected using a Victor X light plate reader (Perkin Elmer Inc., Hopkinton, MA, USA). The levels of each cytosine modification were calculated based on a standard curve generated using the DNA standard provided with the respective kits. All samples were analyzed in duplicate, and positive control DNA, also supplied by the kit manufacturer, was included on each plate to ensure inter-run consistency.

### 4.4. DNA Methylation Analysis of SREBF1, SOCS3, and TXNIP Gene Regions Using the EpiTYPER Assay

The methylation status of specific regions within the SREBF1, SOCS3, and TXNIP genes was assessed using the EpiTYPER assay (Sequenom, San Diego, CA, USA), which evaluates the methylation ratios at targeted CpG sites or clusters, known as CpG units. DNA from PBMCs (500 ng) was subjected to bisulfite conversion using the EZ-96 DNA Methylation Kit (Zymo Research, Tustin, CA, USA) with slight protocol modifications: the CT buffer incubation involved 21 cycles of 15 min at 55 °C and 30 s at 95 °C, and bisulfite-treated DNA was eluted with 100 μL of nuclease-free water.

PCR amplification of the bisulfite-converted DNA was performed using primers specific to the target regions, designed via the EpiDesigner online tool. The following primer sequences were used:

SREBF1: Forward primer (FP) aggaagagagAGGAGGTATAGATTTTGGGTTATGG; Reverse primer (RP) cagtaatacgactcactatagggagaaggctATAAAAAACTCCCTCTTCCAAAAAA

SOCS3: FP aggaagagagGTTTGTTATATTTTGTAGGGAGAGGG; RP cagtaatacgactcactatagggagaaggctACCCAATCTAAAACCAAAAACCTAC

TXNIP: FP aggaagagagAATAGTTTTTGTAATGGAGTGTGGG; RP cagtaatacgactcactatagggagaaggctAAAACAATTACTACTACTTTAAAAACCAAA

Amplicons were processed following the standard EpiTYPER protocol. Each sample was analyzed in duplicate within the experimental setup.

### 4.5. Gene Expression Analysis by Reverse Transcription–Quantitative PCR (RT-qPCR)

Frozen PBMC pellets were thawed on ice and processed for RNA extraction and DNase I digestion using the RNeasy Mini Kit (Qiagen), following the manufacturer’s instructions. RNA concentration, purity, and integrity were assessed via spectrophotometry and agarose gel electrophoresis. Reverse transcription (RT) was performed using the qPCRBIO cDNA Synthesis Kit (PCR Biosystems) with 1 μg of total RNA as the input.

Gene-specific mRNA levels were quantified by quantitative PCR (qPCR) using the equivalent of 30 ng of total RNA subjected to reverse transcription. SREBF1, SOCS3, and TXNIP mRNA levels were measured using SYBR Green-based qPCR with the SYBR Green MasterMix (BioLabs, Ipswich, MA, USA), while IL-6 and MCP-1 mRNA levels were determined using TaqMan qPCR assays with the qPCRBIO Master Mix (PCR Biosystems). Gene expression was quantified using the relative calibrator normalized quantification method, with Hypoxanthine Phosphoribosyltransferase 1 (HPRT1) mRNA as the reference for normalization. An inter-run calibrator sample was included in each plate to account for technical variability across runs and to facilitate the comparison of results between different plates. The calibrator consisted of cDNA synthesized from HEK293T cells. For each experimental qPCR plate, all samples were analyzed in triplicate.

The primer sequences used for SYBR Green-based qPCR assays were as follows: SREBF1 forward primer (FP) ACCAGCGTCTACCATAGCCCT and reverse primer (RP) CATTGAGCAGCCAGACCACT; SOCS3 forward primer GGAGACTTCGATTCGGGACC and reverse primer GGAGCCAGCGTGGATCTG; TXNIP forward primer TGTTCATTCCTGATGGGCGG and reverse primer GCTTTGGGGACCACAATTCG; HPRT1 forward primer TTGGAAAGGGTGTTTATTCCTCA and reverse primer TCCAGCAGGTCAGCAAAGAA. For TaqMan qPCR assays, the following predesigned primers and probes were used: Hs00174131_m1 for IL-6 (Thermo Fisher Scientific, Waltham, MA, USA), Hs00234140_m1 for MCP-1 (Thermo Fisher Scientific), and Hs02800695_m1 for HPRT1 (Thermo Fisher Scientific).

### 4.6. Statistical Analysis and Sample Size Calculation

All the statistical analyses were conducted using SPSS software, version 27.0. Continuous data are presented as the mean ± standard deviation (SD), while categorical data are shown as percentages in both the manuscript and tables. To compare mean values between two independent groups, Student’s *t*-test was used for continuous variables with normal distribution, the Mann–Whitney U test for non-normally distributed variables, and the χ^2^ test for categorical data, as applicable. Comparisons between more than two groups were obtained by ANOVA test. Correlations between variables were evaluated using either Pearson’s or Spearman’s correlation coefficient, according to data distribution. Correlations between FNI and intermediates of demethylation, methylation levels, and mRNA expression were calculated by Pearson’s coefficient, entering FNI as a continuous variable. Variables with skewed distributions were log-transformed before performing the analyses. Two-tailed *p*-value of less than 0.05 was considered statistically significant, with a confidence interval of 95%.

To the best of our knowledge, this is the first study that investigated demethylation intermediates in human PBMCs in relation to the presence of MASLD. For this reason, to evaluate the robustness of our findings, we performed a post hoc sample size calculation starting from the mean 5hmC difference reported between patients with and without MASLD. Thus, we obtained that n = 24 individuals per group were sufficient to obtain results statistically significant with power = 90% and alpha error = 0.05 [40]. The final power of this study was 99.2%, with alpha error = 0.05. 

## Figures and Tables

**Figure 1 ijms-26-01271-f001:**
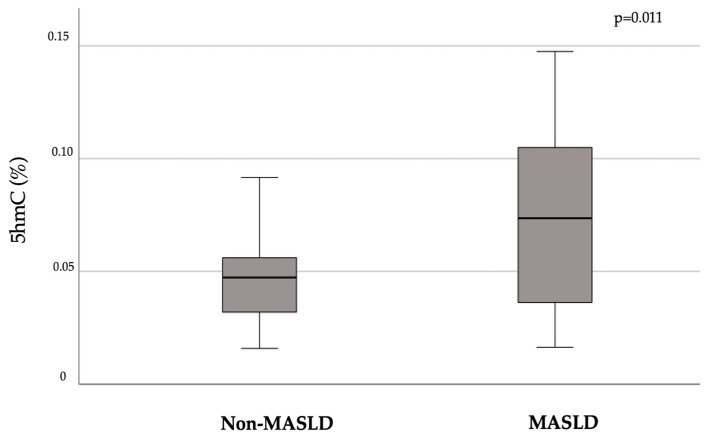
Median (95% C.I.) levels of the demethylation intermediate 5-hydroxymethylcytosine (5hmC) according to the presence of MASLD, identified by hepatic steatosis index (HSI > 36 for MASLD). Student’s *t*-test applied. The data are percentages of signal detected in the samples relative to the DNA standard sample.

**Figure 2 ijms-26-01271-f002:**
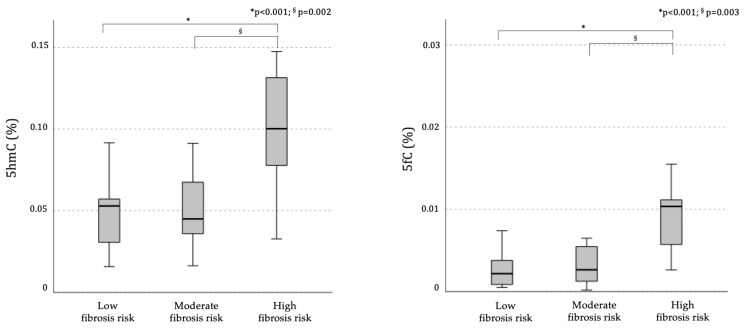
Median (95% C.I.) levels of the demethylation intermediates 5-hydroxymethylcytosine (5hmC) and 5-formylcytosine (5fC) according to the fibrosis risk, considered low: FNI < 0.10, moderate: FNI between 0.10 and 0.33, or high: FNI ≥ 0.33). Bonferroni’s adjusted ANOVA: § high vs. moderate risk; * high vs. low risk. The data are percentages of signal detected in the samples relative to the DNA standard sample.

**Figure 3 ijms-26-01271-f003:**
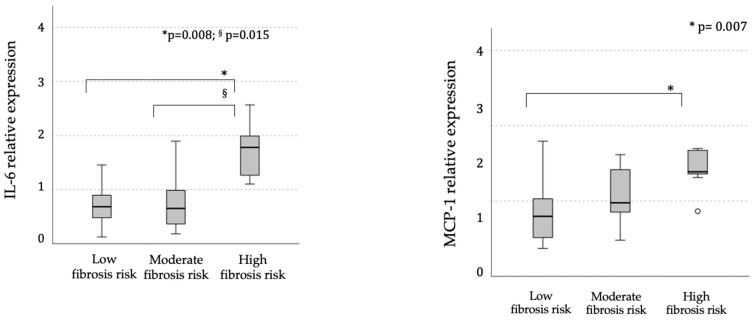
Median (95% C.I.) levels of the IL-6 and MCP-1 mRNA levels in PBMCs of individuals at different estimated risk of liver fibrosis (low: FNI < 0.10, moderate: FNI between 0.10 and 0.33, or high: FNI ≥ 0.33). Bonferroni’s adjusted ANOVA: § high vs. moderate risk; * high vs. low risk.

**Table 1 ijms-26-01271-t001:** Characteristics of study participants in relation to the presence of MALSD, diagnosed by the hepatic steatosis index (HSI). * Mann-Whitney non-parametric test applied.

	MASLD (HSI > 36) n = 56	Non-MASLD (HSI < 36) n = 33	*p*-Value Student’s *t*-Test
Age (years)	62.92 ± 10.95	61.30 ± 9.73	0.49
Sex (M%)	30%	46%	0.16 *
BMI (kg/m²)	31.28 ± 4.30	23.91 ± 1.75	<0.001
Waist circumference (cm)	111.63 ± 10.50	94.80 ± 5.89	0.003
Systolic blood pressure (mmHg)	136.96 ± 19.56	129.07 ± 12.94	0.04
Diastolic blood pressure (mmHg)	82.33 ± 9.81	78.89 ± 8.92	0.13
FBG (mg/dL)	131.08 ± 47.04	107.43 ± 38.27	0.02
HbA1c (%)	7.12 ± 1.62	6.26 ± 1.57	0.02
AST (IU/L)	21.40 ± 8.70	19.17 ± 5.22	0.15
ALT (IU/L)	25.22 ± 12.99	15.80 ± 5.14	<0.001
GGT (IU/L)	27.55 ± 21.39	17.30 ± 8.57	0.019
Total bilirubin (mg/dL)	0.59 ± 0.26	0.59 ± 0.29	0.97
Direct bilirubin (mg/dL)	0.21 ± 0.07	0.25 ± 0.11	0.19
Total cholesterol (mg/dL)	192.19 ± 42.34	200.36 ± 36.81	0.41
HDL (mg/dL)	47.71 ± 15.61	62.40 ± 11.66	<0.001
LDL (mg/dL)	107.77 ± 41.89	102.44 ± 46.92	0.61
Triglycerides (mg/dL)	169.12 ± 87.62	112.69 ± 54.62	0.004
Uric acid (mg/dL)	7.63 ± 10.29	4.63 ± 0.12	0.21
T2DM diagnosis (%)	70%	37%	0.004 *
Lipid-lowering treatments (%)	21%	14%	0.44 *
HSI	43.61 ± 5.75	33.07 ± 1.88	<0.001
FNI	0.23 ± 0.22	0.07 ± 0.07	<0.001

Abbreviations: ALT: alanine aminotransferase; AST: aspartate aminotransferase; BMI: body mass index; FBG: fasting blood glucose; FNI: fibrotic non-alcoholic steatohepatitis (NASH) index; GGT: gamma-glutamyl transferase; HDL: high-density lipoprotein; HbA1c: hemoglobin A1c; HSI: hepatic steatosis index; LDL: low-density lipoprotein.

**Table 2 ijms-26-01271-t002:** Characteristics of the study population according to the estimated risk of liver fibrosis calculated by fibrotic NASH index (FNI; low risk: FNI < 0.10; moderate risk: 0.10 ≤ FNI < 0.33; high risk: FNI ≥ 0.33). * Kruskal-Wallis non-parametric test applied.

	Low Fibrosis Risk (FNI < 0.10) n = 50	Moderate Fibrosis Risk (0.10 ≤ FNI < 0.33) n = 27	High Fibrosis Risk (FNI ≥ 0.33) n = 12	*p*-Value ANOVA
Age (years)	60 ± 9.9	64.9 ± 10.5	62.6 ± 12	0.13
Sex (M%)	50%	79%	64%	0.07 *
BMI (kg/m^2^)	26.6 ±4.2	28.7 ± 4.8	33.7 ± 5.1	<0.001
Waist circumference (cm)	99.8 ±16.4	105.4 ± 11.4	113 ± 10.5	0.26
Systolic blood pressure (mmHg)	132 ± 15.7	137.6 ± 20.4	137.1 ± 19.4	0.42
Diastolic blood pressure (mmHg)	80.9 ± 9.4	81.1 ± 10.2	80.9 ± 10.2	0.99
FBG (mg/dL)	102.9 ± 34	137.6 ± 38.5	160.3 ± 11.6	<0.001
HbA1c (%)	7.1 ± 9.3	7.2 ± 1.2	9.3 ± 1.4	0.60
AST (IU/L)	18.1 ±5.3	20.2 ± 5.3	27.8 ± 12.2	<0.001
ALT (IU/L)	18.4 ± 8.8	21 ± 8.9	34.8 ± 15.8	<0.001
GGT (IU/L)	17.4 ± 6.3	26.8 ± 15.5	47.3 ± 34.3	<0.001
Total bilirubin (mg/dL)	0.50 ± 0.2	0.6 ± 0.3	0.64 ± 0.26	0.30
Direct bilirubin (mg/dL)	0.20 ± 0.08	0.2 ± 0.1	0.23 ± 0.08	0.58
Total cholesterol (mg/dL)	204.6 ± 35.8	182 ± 38.4	176 ± 49.2	0.02
HDL (mg/dL)	61.3 ± 13.9	43.4 ± 7.7	44 ± 12.3	<0.001
LDL (mg/dL)	120.6 ± 29.8	107.4 ± 38.7	97.2 ± 42.2	0.09
Triglycerides (mg/dL)	123.7 ± 62.4	168 ± 90.2	181 ± 83.5	0.018
Uric acid (mg/dL)	5.5 ± 1.3	8.3 ± 11.9	4.9 ± 1.5	0.56
T2DM diagnosis (%)	38%	93%	100%	<0.001 *
Lipid-lowering treatments (%)	12%	22%	25%	0.44
FNI	0.04 ± 0.02	0.18 ± 0.06	0.52 ± 0.17	<0.001 *

Abbreviations: ALT: alanine aminotransferase; AST: aspartate aminotransferase; BMI: body mass index; FBG: fasting blood glucose; FNI: fibrotic NASH index; GGT: gamma-glutamyl transferase; HDL: high-density lipoprotein; HbA1c: hemoglobin A1c; HSI: hepatic steatosis index; LDL: low-density lipoprotein.

## Data Availability

The data presented in this study are available from the corresponding author upon request. The data are not publicly available due to privacy restrictions and lack of specific patient consent.

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
