# Peer review of "Association Between Active DNA Demethylation and Liver Fibrosis in Individuals with Metabolic-Associated Steatotic Liver Disease (MASLD)"

_ijms, 2025, doi:10.3390/ijms26031271_

Round 1
Reviewer 1 Report
Comments and Suggestions for Authors
The manuscript entitled “Association between active DNA demethylation and liver fibrosis in individuals with metabolic-associated steatotic liver disease (MASLD)” explores the role of epigenetic alterations in fibrogenesis and their potential as biomarkers for MASLD progression toward fibrosis. However, the present study lacks relevance and impact, with insufficient and unconvincing data to substantiate the authors' claims. As a result, I recommend the rejection of this paper. However, the following comments may assist the authors in improving their manuscript:
1. The authors did not perform 5fC ELISA for MASLD patients, which was included in the analysis for patients with different fibrosis risk levels. This inconsistency limits the ability to draw comprehensive conclusions about the role of 5fC as a biomarker across the full spectrum of MASLD progression.
2. The results presented in Section 2.1 would benefit from being depicted in a graphical format. Converting the data into visual representations such as graphs, charts, or schematics would enhance clarity and allow readers to grasp trends and relationships within the data better.
3. The authors have stratified patients into MASLD and non-MASLD groups, as well as fibrotic and non-fibrotic categories. However, the manuscript does not include any data specifically addressing fibrosis within the MASLD cohort. This omission is a significant limitation, as it prevents a thorough understanding of the relationship between fibrosis and MASLD in the study population. I recommend providing detailed fibrosis-related data for MASLD patients to strengthen the conclusions and ensure consistency in the stratification analysis.
Author Response
- The authors did not perform 5fC ELISA for MASLD patients, which was included in the analysis for patients with different fibrosis risk levels. This inconsistency limits the ability to draw comprehensive conclusions about the role of 5fC as a biomarker across the full spectrum of MASLD progression.
In our analysis, 5fC was associated with high liver fibrosis risk and did not change significantly in relation to the presence or absence of MASLD. Thus, our study identified 5fC as a specific biomarker of advanced liver damage rather than MASLD itself. This finding aligns with the observations across classes with progressively increased fibrosis risk (see Figure 2). Specifically, patients at low and moderate fibrosis risk exhibited comparable 5fC levels (mean ± SD 5hmC: low risk 0.005 ± 0.007, moderate risk 0.003 ± 0.002), whereas a significant increase in 5fC was observed in the high fibrosis risk subgroup (0.010 ± 0.004; p = 0.007 between groups; p < 0.01 vs low fibrosis risk subgroup). We apologize for not including these findings in the initial version of our manuscript. They have now been added to the Results section: “No difference was observed for 5fC between these subgroups (MASLD mean ± SD 5fC: 0.0055 ± 0.006 vs. non-MASLD: 0.0053 ± 0.004, p = 0.91)” (page 5, lines 141-143) and commented them in the Discussion section: “In our study, 5hmC levels were elevated in the presence of MASLD regardless of fibrosis risk, while 5fC levels showed a progressive increase across fibrosis risk classes. These findings underscore the potential of 5fC as a specific biomarker for identifying patients at high risk of advanced liver fibrosis” (page 8, lines 273-276).
- The results presented in Section 2.1 would benefit from being depicted in a graphical format. Converting the data into visual representations such as graphs, charts, or schematics would enhance clarity and allow readers to grasp trends and relationships within the data better.
Thank you for your suggestion regarding the inclusion of graphical representations for the results in Section 2.1. We agree that visualizing the data would improve clarity and enhance the reader's ability to interpret trends and relationships. To address this, we have provided detailed graphical representations of the data, now included as part of the “Supplementary Materials”. These graphs and charts illustrate the genomic locations of the CpG sites analysed, as well as their methylation levels compared between low and moderate-to-high FNI groups. For further details, we kindly direct you to our response to Comments 4 and 5 from Reviewer 4, where these additions are thoroughly addressed.
- The authors have stratified patients into MASLD and non-MASLD groups, as well as fibrotic and non-fibrotic categories. However, the manuscript does not include any data specifically addressing fibrosis within the MASLD cohort. This omission is a significant limitation, as it prevents a thorough understanding of the relationship between fibrosis and MASLD in the study population. I recommend providing detailed fibrosis-related data for MASLD patients to strengthen the conclusions and ensure consistency in the stratification analysis.
We have now performed new analyses on demethylation pattern in relation to fibrosis within the MASLD cohort, as suggested by the reviewer. The significant association between higher DNA demethylation levels and more advanced fibrosis persisted when the analyses were performed in the subgroup of individuals with MASLD (HSI >36). Among these patients, 64% had moderate to high fibrosis risk; 5hmC and 5fC progressively increased across classes at more pronounced fibrosis risk (5fC (mean ± SD) in low fibrosis risk: 0.0028 ± 0.001, moderate fibrosis risk: 0.0027 ± 0.002, high fibrosis risk: 0.0085 ± 0.003, p = 0.024) and 5hmC (mean ± SD in low fibrosis risk: 0.042 ± 0.023, moderate fibrosis risk: 0.039 ± 0.023, high fibrosis risk: 0.11 ± 0.027, p= 0.002). we thank the reviewer for this important comment and have added these new data in the Results section (page 6, lines 176-183).
Reviewer 2 Report
Comments and Suggestions for Authors
In the current manuscript, the authors investigate the relationship between DNA methylation status and metabolic dysfunction-associated steatotic liver disease (MASLD), with a particular interest on liver fibrosis risk. By measuring global levels of DNA demethylation intermediates—5-hydroxymethylcytosine (5hmC) and 5-formylcytosine (5fC)—in peripheral blood mononuclear cells (PBMCs) from 89 participants, the authors demonstrate that individuals with MASLD and elevated Fibrosis NASH Index (FNI) exhibit significantly increased levels of both 5hmC and 5fC. Additionally, the authors assess the methylation status and mRNA expression of several inflammation- and fibrosis-related genes, including SOCS3, SREBF1, and TXNIP. The results reveal that hypomethylation of the SOCS3 promoter is inversely correlated with FNI and positively associated with the mRNA expression of SOCS3, TXNIP, IL-6, and MCP-1. These findings suggest that active DNA demethylation and gene-specific methylation alterations contribute to inflammation and fibrogenesis in MASLD, potentially serving as biomarkers for liver fibrosis progression.
I have several suggestions and questions for the authors, which I hope will help improve the quality of the manuscript:
1. The material used for the analysis of global methylation levels is peripheral blood cells. How accurately does the methylation status of blood cells reflect that of liver cells? Given that DNA methylation, as an epigenetic mechanism, is highly cell- and tissue-specific, I recommend validating these findings using primary hepatocytes.
2. Please include a brief introduction to epigenetics in gene regulation, highlighting DNA methylation as one of the key mechanisms. Additionally, provide background information on how DNA methylation influences gene expression (activation vs. repression).
3. Fibrosis-related genes should be expanded in the analysis to strengthen the relevance of the findings.
4. Consider adding experimental analyses or discussions on the role of DNA methyltransferases and demethylases in this process, to elucidate their involvement in MASLD progression.
5. In the figures, please include titles and units for the y-axes to improve clarity.
6. Add statistical significance markers (e.g., *) to groups with significant differences in the figures.
Author Response
- The material used for the analysis of global methylation levels is peripheral blood cells. How accurately does the methylation status of blood cells reflect that of liver cells? Given that DNA methylation, as an epigenetic mechanism, is highly cell- and tissue-specific, I recommend validating these findings using primary hepatocytes.
We appreciate the Reviewer’s insightful comment regarding the use of peripheral blood mononuclear cells (PBMCs) for analyzing global methylation levels and the limitations this may introduce. We have revised the discussion in our manuscript to address this important issue: "The interpretation of our results must account for the limitation introduced by using PBMCs, as this approach cannot entirely exclude the potential influence of changes in cell composition on the observed methylation variations. Moreover, while studies such as Johnson et al. [25] have demonstrated that methylation profiles from blood samples can partially capture liver-specific patterns, indicating shared pathways in disease progression, substantial challenges persist in establishing the relevance of our blood-based findings to liver-specific epigenetic changes caused by the disease. Future research should prioritize integrating single-cell approaches or comparing matched primary hepatocytes and blood samples to better delineate tissue-specific epigenetic signatures and their clinical significance" (page 10, lines 346-355).
- Please include a brief introduction to epigenetics in gene regulation, highlighting DNA methylation as one of the key mechanisms. Additionally, provide background information on how DNA methylation influences gene expression (activation vs. repression).
Thank you for your valuable suggestion. We have included a brief introduction to the role of DNA methylation in gene regulation in the revised manuscript. Below is the additional content added to the Introduction section: “The methylation process involves the enzymatic addition of a methyl group pre-dominantly to cytosine residues followed by guanine, forming CpG dinucleotides in DNA and resulting in the formation of 5-methylcytosine (5mC). This modification silences genes by restricting chromatin accessibility and preventing transcription factor binding, thereby inhibiting transcription. DNA methylation is reversible through passive and active demethylation mechanisms […]” (page 2, lines 69-72).
- Fibrosis-related genes should be expanded in the analysis to strengthen the relevance of the findings.
In our study, we primarily focused on inflammation-related genes which could have an impact also in fibrotic processes and that have not been previously studied in human metabolic liver disease. Indeed, SOCS3 suppression was associated to fibroblast activation and progression of fibrosis; studies have also shown that epigenetic inhibition of SOCS3 expression facilitates STAT3 signaling, contributing to experimental fibrosis (new reference 18: Dees C, Pötter S, Zhang Y, et al. TGF-β-induced epigenetic deregulation of SOCS3 facilitates STAT3 signaling to promote fibrosis. J Clin Invest. 2020 May 1;130(5):2347-2363. doi: 10.1172/JCI122462. PMID: 31990678; PMCID: PMC7190914). As for SREBF1, although it is primarily known for its role in lipid metabolism, this gene has been implicated in kidney fibrosis by increasing levels of fibrotic factors such as TGF-β1(new reference 19: Yuan, Q., Tang, B. & Zhang, C. Signaling pathways of chronic kidney diseases, implications for therapeutics. Sig Transduct Target Ther 7, 182, 2022). Similarly, TXNIP plays a crucial role in regulating cellular redox states and has been associated with renal fibrosis linked to aging (new reference 20: He Q, Li Y, Zhang W, Chen J, et al. Role and mechanism of TXNIP in ageing-related renal fibrosis. Mech Ageing Dev. 2021 Jun;196:111475. doi: 10.1016/j.mad.2021.111475. Epub 2021 Mar 26. PMID: 33781783). We do acknowledge that expanding our analysis to include fibrosis-specific genes could have enhanced the relevance of our findings, and have now added a new paragraph commenting on this important point in the Discussion section: “Although these genes are not recognized as primarily fibrosis-specific genes, several recent studies have demonstrated that they centrally take part to fibrosis development and progression in several tissues and organs, such skin, liver and kidney [18-20], and their epigenetic modulation could impact on fibrogenesis in the experimental setting [18]” (page 8, lines 287-291).
- Consider adding experimental analyses or discussions on the role of DNA methyltransferases and demethylases in this process, to elucidate their involvement in MASLD progression.
Thank you for your valuable suggestion. We acknowledge the importance of exploring the role of DNA methyltransferases (DNMTs) and demethylases (TETs) in MASLD progression. In future work, we plan to investigate the specific contributions of DNMTs and TETs, including their regulation and functional impact on 5mC dynamics, to better elucidate their role in MASLD progression. For now, we have included a discussion addressing this aspect in the revised manuscript to highlight its relevance and potential for further exploration: “The origin of the observed methylation changes is difficult to determine, as DNA methylation is influenced by both environmental and genetic factors. Recent studies suggest that epigenetic alterations in MASLD are partially regulated by DNMT and TET enzymes, which are responsible for the deposition and removal of 5mC, respectively [30,35,36]. However, further research is needed to better understand their role in MASLD progression” (page 9, lines 313, 318). New references 30, 35, 36 have been added to the manuscript.
- In the figures, please include titles and units for the y-axes to improve clarity.
Titles and units for the y-axes have been now added to the Figures, as suggested by the reviewer (pages 6 and 7).
- Add statistical significance markers (e.g., *) to groups with significant differences in the figures.
In the previous version of our manuscript, the p-values in the figures derived from the one-way ANOVA test, which evaluates differences across all groups simultaneously rather than between specific pairs. For this reason, we did not include individual significance markers. However, we do agree with the reviewer that data on comparisons between groups would add clarity and robustness to our data and have performed new analyses to test differences between two groups (ANOVA with Bonferroni’s adjustment). Our results showed that patients with high fibrosis risk had significantly higher 5hmC and 5fC levels than both those with low and intermediate fibrosis risk (New Figure 2). Similarly, patients at high fibrosis risk had significantly increased MCP-1 and IL-6 in comparison to those in the low fibrosis risk subgroup. A difference was also found in the comparison between IL-6 levels in high and moderate fibrosis risk (New Figure 3). We thank the reviewer for this comment and have now added new results and figures in the manuscript, along with the appropriate significance markers in the figures (page 6, Figure 2, lines 174-175; page 7, Figure 3, line 227).
Reviewer 3 Report
Comments and Suggestions for Authors
To the authors: The results reported in the manuscript are interesting and could be very useful in the clinic due to the increase in the incidence of the disease. In my opinion, to improve the relevance of your work, you should introduce some changes. The suggestions are due to the non-specialized readers that could access your paper
In the introduction you should mention why you choose the genes, you did it in the discussion, but it could be useful to have the information since the beginning. To avoid an extensive introduction, you could compress the explanation of the importance and process of DNA methylation (lines 60-71 and 77-86).
The results section could be improved by adding more information about the results you presented to give more context to the reader. Tables 1 and 2 need better formatting. In Table 2, in column 1, you state sex M (%), but you gave a rate (25/25,…). In the text, I did not find the citation of reference 10.
It could be useful to include a short explanation about the calculation of FNI index additional to give the references.
Author Response
- In the introduction you should mention why you choose the genes, you did it in the discussion, but it could be useful to have the information since the beginning. To avoid an extensive introduction, you could compress the explanation of the importance and process of DNA methylation (lines 60-71 and 77-86).
Thank you for your insightful feedback. We agree that providing the rationale for the selection of the genes earlier in the manuscript enhances the clarity and flow of the study. Accordingly, we have moved the sentence from the Discussion section to the Introduction: “[…] exploring the methylation profiles and expression of Suppressor of Cytokine Signaling 3 (SOCS3), Sterol Regulatory Element Binding Transcription Factor 1 (SREBF1), and Thioredoxin Interacting Protein (TXNIP), key genes involved in inflammatory processes and liver fibrosis [14-20]”. (page 3, lines 94-97).
- The results section could be improved by adding more information about the results you presented to give more context to the reader. Tables 1 and 2 need better formatting. In Table 2, in column 1, you state sex M (%), but you gave a rate (25/25,…). In the text, I did not find the citation of reference 10.
Thank you for your valuable feedback. We have revised several parts of the results section to improve clarity and make the findings accessible even to readers who are not experts in the field. Additionally, we performed further analyses to enhance the clarity and robustness of the results. The formatting of Tables 1 and 2 has been improved, and we have corrected inaccuracies, including those you pointed out, such as the inconsistency in Table 2 (column 1) and the missing citation of reference 10 in the text. We greatly appreciate your suggestions, which helped us improve the overall quality of the manuscript.
- It could be useful to include a short explanation about the calculation of FNI index additional to give the references.
We thank the reviewer for this comment and have now added a short explanation on the calculation of FNI index in the manuscript: “The risk of liver fibrosis was estimated by calculating the Fibrotic NASH Index (FNI), a recently validated non-invasive scoring system used to assess the risk of advanced liver fibrosis in patients at high risk of Nonalcoholic Steatohepatitis (NASH). FNI was shown to perform better than other non-invasive indexes, such as FIB-4, in identifying liver fibrosis in individuals with metabolic disease and type 2 diabetes [35]. The formula for FNI calculation includes AST, HDL cholesterol, and HbA1c (calculator available online at https://fniscore.github.io/)” (page 11, lines 395-401).
Reviewer 4 Report
Comments and Suggestions for Authors
Barchetta et al. explored the association between active DNA demethylation and liver fibrosis in metabolic-associated steatotic liver disease (MASLD) patients. The data was collected from peripheral PMBC cells. The work has merit, but some revisions and reanalysis are required. I hope my detailed comments below will help the authors improve their work.
Major comments:
1) Based on Tables 1 and 20, the characteristics of the patients and control are not similar. Some differences are obvious due to the disease, but some are not. Controlling for these in the analysis is missing. For example, the effect of sex, controlling for age as a covariate, and so on. Did the authors try to analyze the data separately for males and females? A sex-dependent effect might be a very valuable finding.
2) The authors claimed to explore PMBC; however, there is no report regarding how PMBC was obtained from the blood. Neutrophils are prevalent in blood. They are not PMBC, and no Ficcoll gradient or other method to isolate PMBC is reported. Maybe they measured data from whole blood and not PMBC?
3) Data is poorly presented and not all experimental groups are shown. Figure 1 has non-MASLD vs. MASLD, then in Figure 2 and Figure 3, the control group disappears, and only MASLD patients are shown with data broken into sub-groups by risk; the control group disappeared.
4) Global methylation data is described but with no graph. No significant CpGs data for the SOCS3 is missing, and no data at all for the 2 other genes. All the site-specific DNA methylation data is missing; there are simply no graphs. How can a reader evaluate these findings?
5) The genomic positions of the relevant CpGs are missing, the primers are for bisulfite-converted DNA, and it's nearly impossible to locate their exact position in relation to the gene, the gene promoter, TSS, etc.
6) Many correlations are reported, some of Pearson’s and some of Spearman's (based on the methods), but we don’t know which is which, so we don’t know if the correlations are linear.
Please add scatter plots for these correlations and detailed stat tests so the readers can evaluate the data.
7) EpiTyper is an excellent method, however it is known that its sensitivity is for at least 5% differential methylation (please see the company website about this: https://www.agenabio.com/wp-content/uploads/2015/06/Agena-Bioscience-EpiTyper-Brochure-ONC006302.pdf). cpG 8 at SOCS3 promoter has a smaller difference and therefore this data should be taken with caution and discussed as a limitation.
8) The authors measured methylation in the blood and not in the liver, while an association between these tissues with regard to DNA methylation was reported. This is a weakness of the study that should be addressed in the discussion.
9) Figure 4 and supplementary figures 1 and 2 are mentioned in the Methods section. However, there is no Figure 4 in the manuscript, and no supplementary figures were provided; therefore, I cannot evaluate any of them.
Minor comments:
1) Please spell out NASH the 1st time it appears.
2) Text in the graphs and tables is hard to read. In the version I have, the lines numbering skewed the text. Please fix it.
Author Response
Major comments:
- Based on Tables 1 and 2, the characteristics of the patients and control are not similar. Some differences are obvious due to the disease, but some are not. Controlling for these in the analysis is missing. For example, the effect of sex, controlling for age as a covariate, and so on. Did the authors try to analyze the data separately for males and females? A sex-dependent effect might be a very valuable finding.
As pointed out by the reviewer and reported in the results section, in subjects at moderate-to-high fibrosis risk the prevalence of male sex (74% vs 50%, p= 0.02) and T2DM (95% vs 38%, p<0.001) were significantly higher than in those in the low-FNI group. Thus, we have now performed new analyses controlling for age and for age and T2DM as covariates and found that the association between the levels of the global demethylation intermediates 5hmC and 5fC and FNI persistent statistically significant regardless of i) sex (5hmC: r= 0.64, p<0.001; 5fC: r= 0.61, p<0.001) and ii) sex and presence of T2DM: “At the bivariate analysis, the FNI, considered as a continuous variable, linearly correlated with both 5hmC and 5fC (r= 0.65, p<0.001; r= 0.43, p= 0.016, respectively). These associations persisted statistically significant after adjusting for sex (5hmC: r= 0.64, p<0.001; 5fC: r= 0.61, p<0.001) and for sex and presence of T2DM (5hmC: r= 0.63, p= 0.002; 5fC: r= 0.59, p= 0.003) at the partial correlation analysis” (page 6, lines 192-197). We thank the reviewer for this comment and have now added these new results to the Results section. Conversely, we have recalculated the sex distribution across the 3 subgroups at different fibrosis risk (Table 2) and found no statistically significant difference. We apologize and have now corrected this refuse in the table.
- The authors claimed to explore PMBC; however, there is no report regarding how PMBC was obtained from the blood. Neutrophils are prevalent in blood. They are not PMBC, and no Ficcoll gradient or other method to isolate PMBC is reported. Maybe they measured data from whole blood and not PMBC?
Thank you for pointing out this important aspect. We confirm that our study focused on PBMCs, and their isolation was conducted using a Ficoll-based density gradient centrifugation method. To address this concern, we have updated the Materials and Methods section to include a detailed description of the PBMC isolation procedure in the new paragraph 4.2. Peripheral blood mononuclear cells Isolation. “Peripheral venous blood samples (6 mL) were drawn into EDTA-containing vacutainer tubes (BD). Peripheral blood mononuclear cells (PBMCs) were separated by density gradient centrifugation using the Lymphoprep™ solution (density: 1.077 g/mL; Ax-is-Shield), following the manufacturer's protocol. The isolated cells were either processed immediately or stored as pellets at −80°C for later use” (page 11, lines 402-407).
3) Data is poorly presented and not all experimental groups are shown. Figure 1 has non-MASLD vs. MASLD, then in Figure 2 and Figure 3, the control group disappears, and only MASLD patients are shown with data broken into sub-groups by risk; the control group disappeared.
We apologize if our data presentation was not sufficiently clear and do thank the reviewer for this insightful comment, which gives us the opportunity to clarify our approach. Our study population consisted of 89 subjects and remained unchanged throughout the analyses. Initially, the population was stratified based on the presence or absence of MASLD, as determined by the Hepatic Steatosis Index (HSI). This index is specifically designed to estimate the presence of hepatic steatosis (MASLD) but does not provide information regarding liver fibrosis. Accordingly, Figure 1 illustrates the levels of the demethylation intermediate 5hmC in relation to the presence of MASLD, irrespective of fibrosis status. Subsequently, to address a second research question on the association between liver fibrosis risk and differential demethylation profiles, the same cohort (n=89) was re-stratified based on the risk of liver fibrosis, regardless of the degree of hepatic steatosis. As a result, Figures 2 and 3 present data organized by fibrosis risk groups. We have now revised the Results section and re-organize it in subparagraphs to improve the clarity and accuracy of our data presentation accordingly to this reviewer’s comment: “2.1.2 Demethylation intermediates and liver fibrosis. […] Subsequently, in order to explore the association between differential demethylation profiles and liver fibrosis risk, the same cohort (n=89) was stratified based on the risk of liver fibrosis, estimated by FNI” (page 5, lines 164-169).
4 and 5) Global methylation data is described but with no graph. No significant CpGs data for the SOCS3 is missing, and no data at all for the 2 other genes. All the site-specific DNA methylation data is missing; there are simply no graphs. How can a reader evaluate these findings? The genomic positions of the relevant CpGs are missing, the primers are for bisulfite-converted DNA, and it's nearly impossible to locate their exact position in relation to the gene, the gene promoter, TSS, etc.
We appreciate the Reviewer’s valuable feedback regarding the presentation of the data and its accessibility for evaluation. To address these concerns, we have now included additional data in the Supplementary Materials, which provide detailed information about the analyzed regions and their methylation levels. Specifically, we now provide the methylation data for SOCS3 and the other two genes (SREBF1 and TXNIP) of interest, including graphs that compare methylation levels between low and moderate-to-high FNI groups. We have also added a detailed map in the Supplementary Materials showing the precise genomic locations of the CG sites analyzed, their relation to the gene promoter, transcription start site (TSS), and other relevant regions. This sentence has been added to the Results section: “The profiling was focused on regions previously associated with methylation changes in population studies conducted on whole blood [21,22]. Supplementary Figure 1 illustrates details of the genomic locations of the analyzed CpG sites within the genes, as well as their methylation levels, compared between individuals with low and moderate-to-high FNI scores” (page 7, lines 205-209).
6) Many correlations are reported, some of Pearson’s and some of Spearman's (based on the methods), but we don’t know which is which, so we don’t know if the correlations are linear. Please add scatter plots for these correlations and detailed stat tests so the readers can evaluate the data.
In our study, correlation analyses were conducted to investigate the association between FNI and intermediates of demethylation, methylation levels, and mRNA expression. For all these analyses, Pearson’s coefficients were calculated, and non-normally distributed variables were log-transformed prior to analysis. We apologize if the statistical methods were not clearly described; we have now clarified this point in the Statistics and Results sections: “Correlations between FNI and intermediates of demethylation, methylation levels, and mRNA expression were calculated by Pearson’s coefficient, entering FNI as a continuous variable. Variables with skewed distributions were log-transformed before performing the analyses” (page 12, lines 150-153).
7) EpiTyper is an excellent method, however it is known that its sensitivity is for at least 5% differential methylation (please see the company website about this: https://www.agenabio.com/wp-content/uploads/2015/06/Agena-Bioscience-EpiTyper-Brochure-ONC006302.pdf). cpG 8 at SOCS3 promoter has a smaller difference and therefore this data should be taken with caution and discussed as a limitation.
Thank you for highlighting the sensitivity limitations of the EpiTyper method in detecting smaller differences in methylation. We acknowledge this important point and agree that results for CpG sites with minimal differences, such as CpG8 at the SOCS3 gene, should be interpreted with caution. To address this, we have updated the Discussion section to acknowledge this limitation. The following sentence has been added: "While the EpiTyper method is a robust tool for analyzing DNA methylation, its sensitivity threshold of detecting at least 5% differential methylation must be considered when interpreting results. In particular, the observed methylation differences at CpG8 and CpG13 within the SOCS3 gene fall close to this sensitivity limit. Therefore, these findings should be taken with caution, and additional validation using more sensitive methods would be beneficial to confirm their biological significance" (page 9, lines 308-313).
8) The authors measured methylation in the blood and not in the liver, while an association between these tissues with regard to DNA methylation was reported. This is a weakness of the study that should be addressed in the discussion.
Thank you for pointing out the distinction between measuring DNA methylation in blood versus liver tissue. We have tried to better explain the origin of the DNA on which the methylation level was measured by adding the DNA extraction tissue along the text (PBMCs). We agree that this is a limitation of the study and have addressed this point in the Discussion section. Specifically, we acknowledge the limitations of using PBMCs as a surrogate for liver-specific epigenetic changes, emphasizing the need for future studies to include matched liver and blood samples to validate these findings and explore tissue-specific associations in greater detail, as follows: “The interpretation of our results must account for the limitation introduced by using PBMCs, as this approach cannot entirely exclude the potential influence of changes in cell composition on the observed methylation variations. Moreover, while studies such as Johnson et al. [25] have demonstrated that methylation profiles from blood samples can partially capture liver-specific patterns, indicating shared pathways in disease progression, substantial challenges persist in establishing the relevance of our blood-based findings to liver-specific epigenetic changes caused by the disease. Future research should prioritize integrating single-cell approaches or comparing matched primary hepatocytes and blood samples to better delineate tissue-specific epigenetic signatures and their clinical significance" (page 10, lines 346-355; see also response 1 to Reviewer 2).
9) Figure 4 and supplementary figures 1 and 2 are mentioned in the Methods section. However, there is no Figure 4 in the manuscript, and no supplementary figures were provided; therefore, I cannot evaluate any of them.
We apologize for the oversight in the submission of figures. This was an error on our part during the file submission process. We have now revised and recreated those figures, ensuring their accuracy and clarity. The updated material in now included in Supplementary Figure 1 of the revised manuscript. We hope these additions address your concerns and allow for a thorough evaluation of our findings. For further details, we kindly refer you to our Response to your comments 4 and 5.
Minor comments:
- Please spell out NASH the 1st time it appears.
We thank the reviewer for this comment and have now spelt NASH out the first time this acronym was mentioned in the abstract and manuscript.
- Text in the graphs and tables is hard to read. In the version I have, the lines numbering skewed the text. Please fix it.
We apologize for this inconvenience and reformatted the graphs and tables in this version of the manuscript to improve readability.
Round 2
Reviewer 4 Report
Comments and Suggestions for Authors
The authors answered most of my comments and questions; thanks.
- One minor comment is that since many correlations are reported, adding graphs (ie, scatter plots) with the data of these correlations, at least the significant ones, will greatly improve the quality of the paper (can be as supplemental figs).
Author Response
The authors answered most of my comments and questions; thanks.
- One minor comment is that since many correlations are reported, adding graphs (ie, scatter plots) with the data of these correlations, at least the significant ones, will greatly improve the quality of the paper (can be as supplemental figs).
Thank you for your valuable feedback and constructive suggestions. We appreciate your comment regarding the addition of scatter plots for the significant correlations. As per your suggestion, we have generated the requested graphs, focusing on the DNA methylation data. These scatter plots illustrate the significant correlations and are included as supplemental figures. We hope these additions meet your expectations and contribute.